# Evaluating soluble Axl as a biomarker for glioblastoma: A pilot study

Daniel Raymond[1]☉, Melanie Fukui [ID]²¤ᵃ, Samuel Zwernik³¤ᵇ, Amin Kassam²¤ᵃ, Richard Rovin [ID]²☉*, Parvez Akhtar³☉

1 Department of Biology, Northern Michigan University, Marquette, Michigan, United States of America,
2 Aurora Neuroscience Innovation Institute, Milwaukee, Wisconsin, United States of America, 3 Advocate Aurora Research Institute, Milwaukee, Wisconsin, United States of America

☉ These authors contributed equally to this work.
¤a Current address: Department of Neurosciences, Northwest Community Hospital, Arlington Heights, Illinois, United States of America
¤b Current address: Department of Surgery, Henry Ford Health System, Detroit, Michigan, United States of America
* richard.rovin@aah.org

**Data Availability Statement:** All datasets are available from the Dryad database: Rovin, Richard (Forthcoming 2024). Soluble Axl Biomarker Datasets [Dataset]. Dryad. https://doi.org/10.5061/dryad.wpzgmsbwp.

## Abstract

With current imaging, discriminating tumor progression from treatment effect following immunotherapy or oncolytic virotherapy of glioblastoma (GBM) is challenging. A blood based diagnostic biomarker would therefore be helpful. Axl is a receptor tyrosine kinase that is highly expressed by many cancers including GBM. Axl expression is regulated through enzymatic cleavage of its extracellular domain. The resulting fragment can be detected in serum as soluble Axl (sAxl). sAxl levels can distinguish patients with melanoma, hepatocellular carcinoma, and pancreatic ductal adenocarcinoma from healthy controls. This is a pilot study to determine if sAxl is a candidate biomarker for GBM. The sAxl levels in the serum of 40 healthy volunteers and 20 GBM patients were determined using an enzyme-linked immunosorbent assay (ELISA). Pre- and post- operative sAxl levels were obtained. Volumetric MRI evaluation provided GBM tumor volume metrics. There was no significant difference in the sAxl levels of the volunteers (30.16±1.88 ng/ml) and GBM patients (30.74±1.96 ng/ml) p = 0.27. The postoperative sAxl levels were significantly higher than preoperative levels (32.32±2.26 ng/ml vs 30.74±1.96 ng/ml, p = 0.03). We found no correlation between tumor volume and sAxl levels. Axl expression was low or absent in 6 of 11 (55%) patient derived GBM cell lines. Given the wide range of Axl expression by GBM tumors, sAxl may not be a reliable indicator of GBM. However, given the small sample size in this study, a larger study may be considered.

## Introduction

Since 1977, interval imaging to monitor a brain tumor's response to treatment has been standard of care [1]. However, distinguishing true progression from treatment effect (pseudoprogression) is challenging, particularly so with the advent of immunotherapy and oncolytic virotherapy [2].

**Funding:** The author(s) received no specific funding for this work.

**Competing interests:** The authors have declared that no competing interests exist.

A circulating biomarker that reflects the biological activity of the tumor would be useful. While available for breast cancer [3], lung cancer [4], melanoma [5], prostate cancer [6], and colorectal cancer [7], a circulating biomarker to monitor the course of gliomas remains elusive.

In earlier work, we found that patient derived glioblastoma (GBM) cell lines are susceptible to productive Zika virus infection especially when Axl is overexpressed [8]. Axl is a member of the TAM family of receptor tyrosine kinases (RTKs) along with Tyro3 and Mer. Like other RTKs, Axl has an extracellular domain for ligand binding, a transmembrane domain, and an intracellular domain. When stimulated through its ligand, growth-arrest specific factor 6 (Gas6), Axl activates myriad intracellular signaling pathways that contribute to the cancer phenotype, including: epithelial-mesenchymal transition, survival, proliferation, angiogenesis, chemotherapy resistance, and immune suppression [9, 10] (Fig 1, right).

Soluble Axl (sAxl) is the byproduct of regulation of Axl expression through post-translational deactivation by enzymatic cleavage. The sheddases ADAM10 and ADAM17 cleave the extracellular domain of Axl, and this product can be identified in the bloodstream as sAxl [9] (Fig 1, left). Many cancers highly express Axl [10]. In these cancers, cleavage of the Axl ectodomain leads to high levels of circulating sAxl. In this way, sAxl can serve as a biomarker for hepatocellular carcinoma [11], pancreatic cancer [12], and melanoma [13].

Given that Axl is also overexpressed in GBM [14], we hypothesized that serum sAxl levels should be elevated and could serve as a biomarker as it does in other solid tumors. Therefore, we designed this pilot study to determine if circulating sAxl levels are elevated in patients with GBM compared to healthy controls and to determine if there is a relationship between tumor volume and sAxl levels.

## Materials and methods

This study was approved by the Northern Michigan University Institutional Review Board #HS19-1033 and the Aurora St Luke's Institutional Review Board #14–79. Participants signed a written informed consent document before enrolling in this study.

### Study design

This is a clinico-pathological correlation study using biospecimens (blood and tumor tissue) collected per an existing Aurora St Luke's Medical Center biorepository protocol (Prospective Biospecimen Collection, Storage, and Distribution).

### Study populations

Control cohort. Volunteers without pre-existing medical conditions were eligible to participate. The control group included students and faculty at Northern Michigan University, Marquette, Michigan. Student volunteers were recruited from the CLS 100 Phlebotomy course. Their blood draws for soluble Axl analysis were part of their standard course work. Blood collection for the control cohort took place between April 8th and May 6th, 2019.

Glioblastoma patients. Patients with preoperative imaging consistent with GBM were eligible to participate in this study. Patients underwent medically indicated surgery at Aurora St Luke's Medical Center, Milwaukee, Wisconsin. Blood was collected before and after surgery and did not necessitate additional venipuncture. Blood collection for the GBM cohort took place between April 1st, 2018 and October 31st, 2018.

Patient derived cell lines. The patient derived GBM cell lines used to determine Axl expression and sAxl levels in cell culture supernatant were previously established. They are not derived from patients participating in the current study.

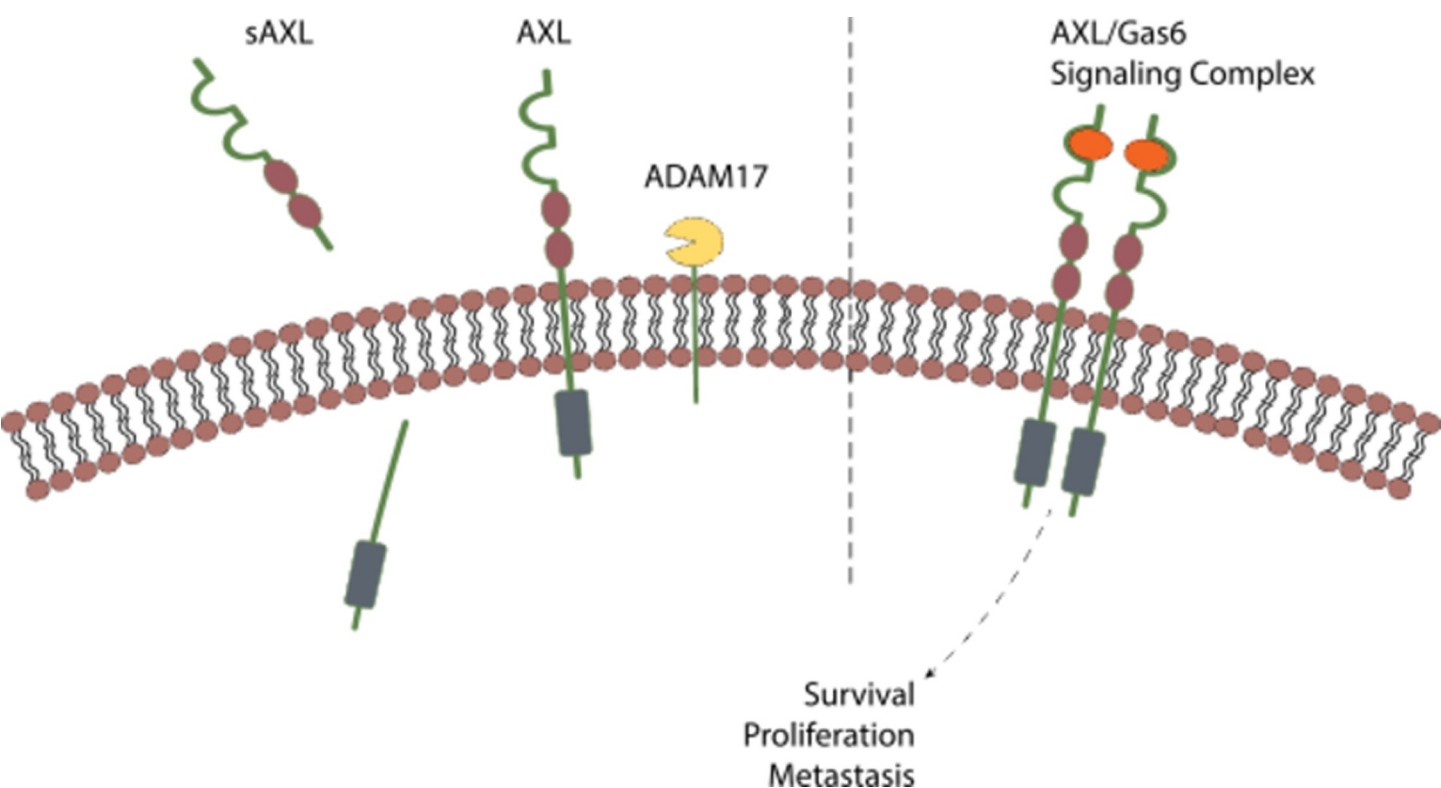

**Fig 1. Right. Axl signaling pathways and post-translational cleavage**. Axl is a transmembrane receptor tyrosine kinase. Gas-6 ligand binding leads to dimerization and activation of cellular pathways including: epithelial-mesenchymal transition, survival, proliferation, angiogenesis, chemotherapy resistance, and immune suppression. left. Axl signaling pathways and post-translational cleavage. One mechanism to regulate Axl expression is cleavage of the extracellular domain by the sheddases ADAM10 and ADAM17. The cleavage product makes its way into the bloodstream as soluble Axl (sAxl).

### Serum collection

The blood collection tubes were labeled with a randomly generated ID number. The samples were deidentified except for age, biological sex, cohort, and, for the GBM cohort, the time period (pre- or post- operative) of the blood draw.

Serum was separated from whole blood samples using gold-top serum separator tubes, which were then centrifuged at 1000 rpm for 5–10 minutes. Serum was aliquoted and stored at -80° C.

### Enzyme-linked immunosorbent assay for sAxl concentration

Serum sAxl and cell culture supernatant sAxl concentrations were determined by enzyme linked immunosorbent assay (ELISA) using the commercially available human Axl DuoSet ELISA kit (R&D Biosystems, Minneapolis, MN) with previously reported optimizations [15]. The 96-well ELISA plates were read in a multi-mode microplate reader (BioTek Synergy H1, Agilent Technologies, Santa Clara, CA). The optical density was measured at 450nm and 540nm. For wavelength correction, the 540nm measurement was subtracted from the corresponding 450nm value. After adjusting for background by normalizing against blank wells, a four-parameter logarithmic curve-fit was generated in GraphPad Prism. A standard curve using known sAxl standards was generated and used to interpolate the sAxl concentration of the samples. For each plate, standards were applied in duplicate, and samples were applied in triplicate. The assay was repeated either the same day using a duplicate 96-well plate on a separate day.

## MRI volumetric tumor analysis

MRI scans were analyzed by a board certified neuroradiologist (M.B.F.) using BrightMatter Plan software (Synaptive Medical, Toronto, Canada). From gadolinium enhanced T-1 weighted images, a 3D model of the tumor was generated. From the pre-operative scans, total tumor volume, volume of enhancing tumor, and volume of necrotic tumor were determined. The ratios of enhancing tumor volume to total tumor volume, necrotic tumor volume to total tumor volume, and necrotic tumor volume to enhancing tumor volume were calculated.

## Cell lines and cultures

Previously established and characterized patient derived glioblastoma stem cell lines were maintained in NeuroCult NS-A basal medium (Stemcell Technologies), supplemented with B-27 without vitamin A, N-2, GlutaMAX and Pen/Strep (Thermo Fisher Scientific), BSA and heparin (Sigma-Aldrich), human recombinant bFGF and EGF (20 ng/ml each; PeproTech Inc.). All cell lines were routinely tested for mycoplasma contamination by using MycoAlert Detection Kit (Lonza Inc.).

## Western blotting

Patient derived GBM stem cells were collected and lysed in RIPA buffer with complete protease inhibitor cocktail (Roche). Lysates were resolved by 4–12% SDS/PAGE and electrotransferred to nitrocellulose iBlot 2 Transfer Stacks (Life Technologies, IB23002). Membranes were blocked with 5% nonfat dry milk in 1x TBS and incubated overnight at 4˚C with anti-AXL primary antibody (1: 1000 dilution, R&D Systems, AF154) or β-actin antibody (Sigma, A2066), and subsequently incubated for 1 hour at room temperature with HRP-coupled secondary antibody. All membranes were scanned using the Odyssey infrared imaging system (LI-COR Biosciences) in conjunction with the Clarity Western ECL Substrate (Bio-Rad).

## RNA extraction and qRT-PCR

Total RNA was extracted from patient derived GBM stem cells using the RNeasy Mini Kit (Qiagen). Isolated RNAs (1 μg total RNA) were then digested with 1 unit of DNase I (NEB) at 37˚C for 25 min to remove genomic DNA contamination before being processed for reverse transcription. Quantitative real time polymerase chain reaction (qRT-PCR) was performed using iTaq Universal SYBR Green One-Step Kit (Bio-Rad, 1725151) according to the manufacturer's instructions on a Roche LightCycler 480 instrument. The primers for Axl quantification were Forward primer 5′–GGTGGCTGTGAAGACGATGA–3′; Reverse primer 5′–CTCAGATAC TCCATGCCACT–3′. The control primers were GAPDH Forward primer 5′– GGATTTGGT CGTATTGGG –3′; Reverse primer 5′– GGAAGATGGTGATGGGATT –3′. Relative expression quantification was performed based on the comparative CT Method ($2^{-\Delta\Delta Ct}$), using GAPDH as an endogenous reference control.

## Statistical analysis

Data from ELISAs were interpreted using a four-parameter logarithmic curve in GraphPad Prism version 10.1.0 (GraphPad Software, San Diego, CA). Determination of statistically significant variation between the two groups was performed using a two-tailed, unpaired student's t-test. For datasets with three or more groups were analyzed using one-way ANOVA with Dunnett's post-hoc multiple comparisons test. Graphical representation of datasets was performed in GraphPad with error bars representing either 95% confidence intervals or mean ± SEM. Correlation between tumor volume measurements and sAxl levels was determined using the

Spearman method; p values are provided to judge significance. This and graphical representation of datasets were performed using Stata version 15 (StataCorp, College Station, TX).

## Results

### Baseline sAxl levels in the control and GBM populations

There were 40 volunteers in the control group and 20 patients in the GBM group. The sAxl level for the healthy controls of $30.16 \pm 1.88$ ng/ml (mean $\pm$ SD) and the preoperative sAxl level in the GBM cohort of $30.74 \pm 1.96$ ng/ml were not significantly different, p = 0.27. (Table 1)

### sAxl levels in response to surgery

There were 19 GBM patients with matched pre- and post- operative sAxl levels. We found that the post-operative sAxl level ($32.32 \pm 2.26$ ng/ml) was significantly higher than the pre-operative sAxl level ($30.74 \pm 1.96$ ng/ml), p = 0.03. In 11 (58%) patients, the preoperative sAxl level was greater than the postoperative level. (Fig 2)

### Correlation between preoperative tumor volume and sAxl levels

Twenty patients had matched preoperative tumor volumes measurements and sAxl levels. There was no correlation between total, enhancing, and necrotic tumor volumes and sAxl level. Nor was there a correlation between necrotic to enhancing, necrotic to total, or enhancing to total volume ratios and sAxl levels. (Table 2 and Fig 3)

### Axl expression and sAxl levels in patient derived glioblastoma cell lines

Tumor Axl expression from the GBM patients participating in this study was not determined. Therefore, to get a sense of Axl expression and sAxl levels in GBM, we performed Western blot (Fig 4A) and qRT-PCR (Fig 4B) using our previously established GBM cell lines. The original uncropped, unadjusted gel/blot images are compiled in the PDF document named S1 Raw images. This file is available as Supporting Information. We also determined sAxl levels in the cell culture supernatants using ELISA. No or low Axl expression was seen in 55% of cell lines by Western blot and 33% of cell lines by qRT-PCR. The sAxl levels in the supernatants were not detected or low in 40% of cell lines. (Table 3)

Relative Axl levels are reported as mean (standard deviation) as determined by qRT-PCR. This is visualized in Fig 4. sAxl levels in the cell culture supernatant as determined by ELISA.

## Discussion

Soluble Axl (sAxl) has emerged as a promising biomarker for the early detection, diagnosis, and monitoring of various cancers [10], including pancreatic ductal adenocarcinoma [12], hepatocellular carcinoma [11, 15], and melanoma [13]. We hypothesized that given the increased expression of Axl in GBM [14], circulating sAxl levels would correspondingly be

**Table 1. Characteristics of the healthy control and glioblastoma (GBM) cohorts.**

|  | Control (N = 40) | GBM (N = 20) | p value |
|---|---|---|---|
| Age (years) | 30.7±16.7 | 64.2±10.8 | = 0.000 |
| Female | 27 (67.5%) | 8 (40%) |  |
| sAxl (ng/ml) | 30.16±1.88 | 30.74±1.96 | = 0.27 |

sAxl: soluble Axl

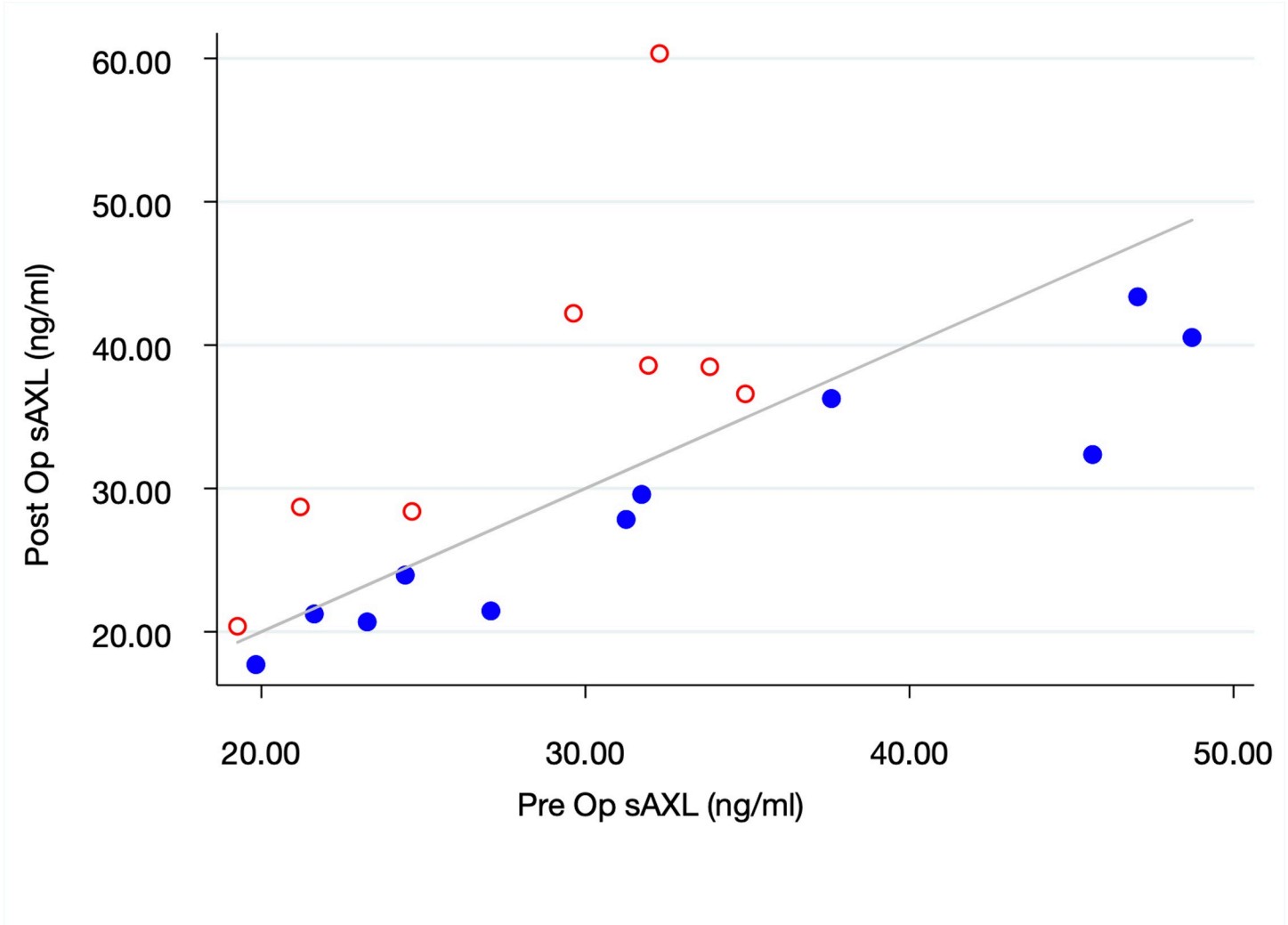

**Fig 2. The relationship between pre- and post-operative soluble Axl levels.** The blue closed circles represent patients with postoperative sAxl levels lower than their preoperative levels. The red open circles represent patients with postoperative sAxl levels higher than preoperative. The diagonal line denotes pre- and post-operative sAxl equality.

elevated. We designed this pilot study to determine if sAxl levels were elevated in patients with GBM and if levels correlated with tumor burden. Though the sAxl levels in our GBM cohort were comparable to levels reported in the literature for other cancers, we did not find a difference in sAxl levels between healthy controls and GBM patients. (Table 4) Nor did we find a correlation between sAxl levels and tumor volume. (Table 2 and Fig 3).

**Table 2. Correlation between preoperative MRI tumor volume measurements and sAXL level.**

| | Total Volume | Enhancing Volume | Necrotic Volume | Necrotic to Enhancing volume | Necrotic to Total Volume | Enhancing to Total Volume |
|---|---|---|---|---|---|---|
| Spearman correlation coefficient | -0.2279 | -0.2286 | -0.1083 | 0.1053 | 0.1053 | -0.1053 |
| Observations | 20 | 20 | 20 | 20 | 20 | 20 |
| p value | 0.3338 | 0.3324 | 0.6494 | 0.6587 | 0.6587 | 0.6587 |

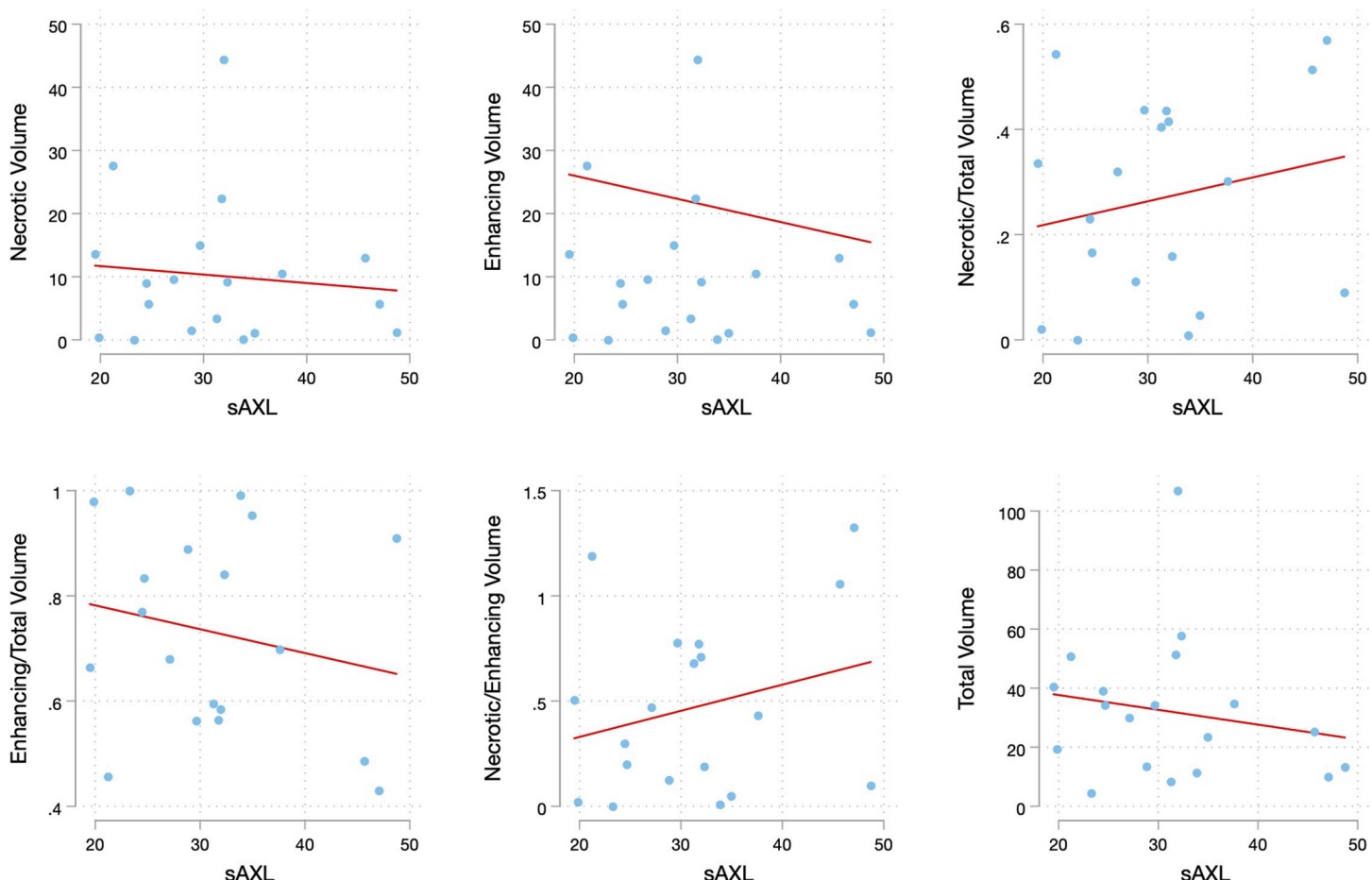

**Fig 3. The relationship between tumor volume metrics and soluble Axl levels.** Multiple scatter plots visualize the lack of correlation between tumor volume metrics and sAxl levels. The total tumor volume, necrotic tumor volume, and enhancing tumor volume were determined through volumetric analysis of preoperative MRI scans. The ratios of necrotic to total volume, enhancing to total volume, and necrotic to enhancing volume were then calculated.

As noted in Table 3 above, we found a range of Axl expression in our established GBM cell lines, including no or low Axl expression in just over half of the specimens by Western blot. Moreover, cell lines with no or low Axl expression had no or low sAxl in the cell culture supernatant. A close review of Fig 1A from the Hutterer et al., paper showed 16 of 30 GBM samples had no or low Axl mRNA expression [14].

The regulation of Axl expression is complex and involves transcriptional, post-transcriptional, and post-translational pathways [10]. The post-translational cleavage of the Axl extracellular domain by ADAM10 and ADAM17 creates the soluble Axl fragments [9, 18]. It is possible, then, that in our small cohort, no or low Axl expressing GBMs were overrepresented, lowering the mean sAxl level.

We further hypothesized that sAxl levels would decline following cytoreduction. Unexpectedly, we found postoperative sAxl levels were significantly higher than preoperative. The reason for this is unclear.

In the hope of using sAxl as a proxy for tumor burden [19], we determined total tumor volume, volume of enhancing (and presumably viable) tumor, and volume of necrotic tumor. We found no correlation between these tumor metrics and sAxl levels.

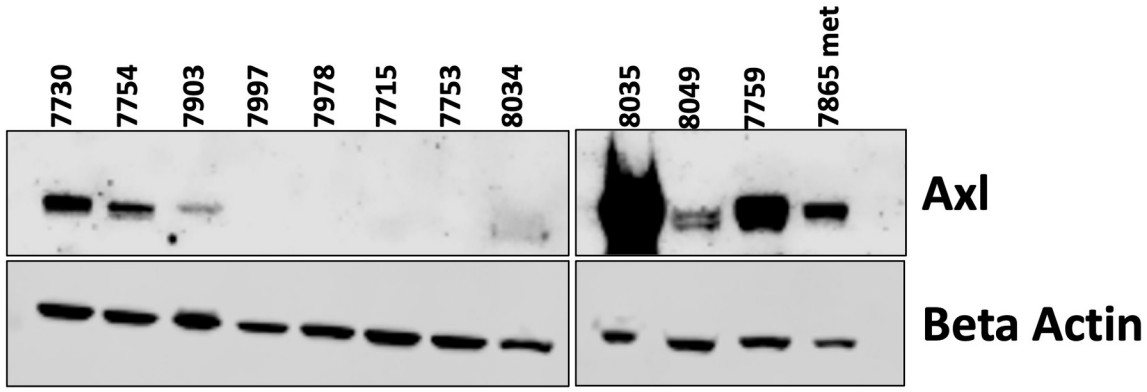

**Fig 4. The expression of Axl and soluble Axl in patient derived glioblastoma cell lines.** (A) Axl protein expression as determined by Western blot. (B) Axl mRNA expression as determined by qRT-PCR. The mean value and standard deviations are plotted. Each measurement was performed in triplicate and the experiment repeated at least twice.

Though our findings that sAxl levels in GBM do not discriminate healthy controls from patients with tumor and do not reflect tumor burden are different from findings in other cancers, they are not unique. Gustafsson et al., found that sAxl levels in patients with renal cell carcinoma (RCC) were lower than in healthy controls, did not correlate with tumor Axl

**Table 3. Axl expression and sAxl levels in patient derived glioblastoma cell lines.**

| Cell Line | Western Blot | Axl Level | sAxl (ng/ml) |
|---|---|---|---|
| 7715 | No | | |
| 7997 | No | 1.00 (0.00) | 0 |
| 7753 | No | 2.92 (0.49) | 0 |
| 7754 | Med | 14.73 (3.63) | 4.73 |
| 7865 | | 41.09 (12.86) | |
| 7714 | | 43.16 (10.07) | 3.24 |
| 7730 | Med | 58.06 (11.83) | 5.89 |
| 7978 | No | 101.26 (20.63) | 0 |
| 8034 | Low | 137.15 (30.57) | 0.71 |
| 8035 | High | 152.99 (31.45) | 7.73 |
| 7759 | High | 154.74 (32.86) | 3.17 |
| 8049 | Med | 187.41 (57.80) | 1.92 |

Western Blot data were qualitatively categorized as No, Low, Medium, or High.

**Table 4. Serum levels of soluble Axl in several cancers.**

| Cohort | Number | sAXL (ng/ml) | p value | Reference |
|---|---|---|---|---|
| **Pancreas** | | | | Martinez-Bosch et al., 2021 [12] |
| Hmar cohort | | Median (IQR) | | |
| Healthy Control | 7 | 39.45 (13.03) | 0.002 | |
| Chr Pancreatitis | 21 | 44.82 (21.75) | 0.003 | |
| PDAC | 31 | 59.78 (25.38) | ref | |
| HClinic cohort | | | | |
| Healthy Control | 46 | 40.03 (14.13) | <0.0001 | |
| Chr Pancreatitis | 16 | 36.34 (11.05) | <0.0001 | |
| PDAC | 80 | 52.66 (30.08) | ref | |
| **Liver Cancer** | | Median | | Reichl et al., 2015 [16] |
| HCC | 311 | 18.56 | <0.0001 | |
| Healthy Control | 237 | 13.39 | ref | |
| Cirrhosis | 60 | 12.17 | 0.99 | |
| **Liver Cancer** | | Median (IQR) | | Dengler et al., 2017 [15] |
| HCC | 347 | 78.69 (55.09–101.5) | ≤0.001 | |
| Healthy Control | 75 | 40.15 (35.22–46.85) | | |
| Cirrhosis | 155 | 94.74 (66.38–132.5) | | |
| **Liver Cancer** | | | | Song et al., 2020 [17] |
| HCC | 80 | 202.0 (154.6–252.6) | <0.001 | |
| Healthy Control | 80 | 67.8 (46.9–89.3) | | |
| Cirrhosis | 80 | 150.5 (100.4–191.1) | | |
| **Liver Cancer** | | Median | <0.05 | Fu et al., 2022 [11] |
| HCC | 190 | 33.55 | | |
| Healthy Control | 82 | 11.39 | | |
| Cirrhosis | 128 | 29.98 | | |
| **Melanoma** | | Mean, 95% CI | <0.0001 | Flem-Karlsen et al., 2020 [13] |
| Stage III | 160 | 26.6, 24.3–28.9 | | |
| Stage IV | 50 | 54.1, 50.7–57.6 | | |
| **GBM** | | Mean ± SD | 0.27 | Current Study |
| Patient | 20 | 30.74±1.96 | | |
| Healthy Control | 40 | 30.16±1.88 | | |

The serum levels of soluble Axl from patients in the current study (glioblastoma and healthy control cohorts) are compared to those reported for patients with cirrhosis, hepatocellular carcinoma (HCC), chronic pancreatitis (Chr Pancreatitis), pancreatic ductal adenocarcinoma (PDAC), and melanoma. The literature references are provided.

expression, nor tumor size [20]. This suggests that tumor type and its microenvironment impact Axl regulation and sAxl production.

## Conclusion

This study is innovative because we additionally included a comparison of pre- and post- operative sAxl levels, and we correlated volumetric MRI tumor metrics with sAxl levels. While this study does not support sAxl as a biomarker for GBM, it is not fair to make a definitive statement given our small sample size. As identifying a GBM biomarker for detection, monitoring, and prognostication, is important, a larger study can be considered.

## Supporting information

**S1 Raw images. The original uncropped, unadjusted gel/blot images.**
(PDF)

## Acknowledgments

The authors thank the Aurora St Luke's BSRC department for biospecimen management and deidentification of MRI scans.

## Author Contributions

**Conceptualization:** Daniel Raymond, Richard Rovin, Parvez Akhtar.

**Data curation:** Daniel Raymond, Richard Rovin, Parvez Akhtar.

**Formal analysis:** Daniel Raymond, Melanie Fukui, Richard Rovin, Parvez Akhtar.

**Investigation:** Parvez Akhtar.

**Methodology:** Daniel Raymond, Melanie Fukui, Samuel Zwernik, Richard Rovin, Parvez Akhtar.

**Project administration:** Daniel Raymond, Richard Rovin, Parvez Akhtar.

**Resources:** Amin Kassam.

**Validation:** Richard Rovin, Parvez Akhtar.

**Visualization:** Richard Rovin.

**Writing – original draft:** Richard Rovin, Parvez Akhtar.

**Writing – review & editing:** Daniel Raymond, Richard Rovin, Parvez Akhtar.

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
