## [Decision Letter · Decision Letter 0]

23 Apr 2024

PONE-D-24-10611Evaluating soluble Axl as a biomarker for glioblastoma: a pilot studyPLOS ONE

Dear Dr. Rovin,

Thank you for submitting your manuscript to PLOS ONE. After careful consideration, we feel that it has merit but does not fully meet PLOS ONE’s publication criteria as it currently stands. Therefore, we invite you to submit a revised version of the manuscript that addresses the points raised during the review process.

A number of issues were identified by both reviewers; particularly Reviewer #2.  These should be addressed if the authors plan to submit a revised manuscript to the journal.

We look forward to receiving your revised manuscript.

Kind regards,

Salvatore V Pizzo

Academic Editor

PLOS ONE

Journal Requirements:

Additional Editor Comments:

A number of issues were identified by both reviewers; particularly Reviewer #2. These should be addressed if the authors plan to submit a revised manuscript to the journal.

Reviewers' comments:

Reviewer's Responses to Questions

**Comments to the Author**

1. Is the manuscript technically sound, and do the data support the conclusions?

Reviewer #1: Yes

Reviewer #2: Partly

2. Has the statistical analysis been performed appropriately and rigorously? 

Reviewer #1: No

Reviewer #2: Yes

3. Have the authors made all data underlying the findings in their manuscript fully available?

Reviewer #1: No

Reviewer #2: Yes

4. Is the manuscript presented in an intelligible fashion and written in standard English?

Reviewer #1: Yes

Reviewer #2: Yes

5. Review Comments to the Author

Reviewer #1: The pilot study conducted by Raymond et al. aimed to assess the potential of soluble Axl (sAxl) as a biomarker candidate for glioblastoma (GBM). Previous research has indicated frequent expression of the AXL receptor and its putative ligand, Gas6, in human gliomas. Notably, both Axl and Gas6 are often overexpressed in glioma and vascular cells, correlating with a poor prognosis in GBM patients (Hutterer et al., reference 14). However, the study found no significant difference in sAXL concentration between GBM patients and a healthy control group.

When examining glioblastoma cell lines, the authors observed significant AXL expression in 45% of the samples, consistent with previous findings. Despite this, the study's results suggest that serum sAXL concentration alone may not serve as a reliable diagnostic tool for glioblastoma.

Given that Hutterer et al. reported a higher number of GBM specimens with elevated GAS6 levels, it would be worthwhile for the authors to consider including Gas6 measurements in their sample analysis. Additionally, although glioblastomas release tumoral content into the bloodstream, and certain biomarkers are known to traverse the blood-brain barrier, it's plausible that some molecules may be retained in the cerebrospinal fluid. Therefore, assessing sAXL levels in cerebrospinal fluid samples could provide valuable insights and should be considered by the authors.

Furthermore, expanding Table 3 to incorporate data from additional studies, such as reference 18 and various studies on hepatocellular carcinoma, would enrich the comparative analysis.

10.4143/crt.2019.749

10.18632/oncotarget.17598

10.1002/ijc.29394

Line 95. “1019”.

Line 187, line 188, line 191, line 219. The correlation of between magnitudes should be quantified and included in the results.

The values of the qRT-PCR analysis should be included in the study, not only the bar. The type of magnitude should be included in the figure legend (mean or median, SEM or SD) with the number of measurements performed.

Reviewer #2: In the manuscript “Evaluating soluble Axl as a biomarker for glioblastoma: a pilot study”，The author found that although AXL is highly expressed in many cancers, including glioblastoma, the byproduct of AXL expression, sAXL, cannot serve as a reliable biomarker for diagnosing glioblastoma. This is a carefully done study and the findings are of considerable interest. A few minor revisions are list below.

1. Results – It would be helpful to the reader if the Figures can be improved with high quality and pixels.

2. Tables – The table looks a little bit complicated, just have a try the three-line table.

3. Does this imply that the experimental results may be meaningless as there is no difference in preoperative AXL levels between the control group and the GBM group?

4. By searching the keyword “glioblastoma” and “AXL”, it is not hard to find other articles on the connection between them. Please describe the difference between this manuscript and other articles and point out the innovations.

5. The language and syntax and format need more attention and the authors might benefit from the assistance of an English language editor.

Suggestion：

1. Use high quality pictures.

2. Identify the innovative points of the article.

3. Expand the sample size and re-study to reduce false-negative results.

4. Some of the references cited in the article are outdated.

6. PLOS authors have the option to publish the peer review history of their article (what does this mean?). If published, this will include your full peer review and any attached files.

Reviewer #1: No

Reviewer #2: No

---

## [Author Response · Author response to Decision Letter 0]

1 Jun 2024

May 20, 2024

To: Salvatore V Pizzo, Academic Editor, PLOS ONE 

Re: Revision of PONE-D-24-10611 Evaluating soluble Axl as a biomarker for glioblastoma: a pilot study

Thank you for the careful review of our manuscript. We appreciate the opportunity to respond to comments and improve our submission. 

Regarding administrative and general editorial comments:

1. We reviewed PLOS ONE’s style requirements and file naming conventions. 

2. Our full ethics statement appears in the Methods section, lines 83-85. I reproduce it here: “This study was approved by the Northern Michigan University Institutional Review Board #HS19-1033 and the Aurora St Luke’s Institutional Review Board #14-79. Participants signed a written informed consent document before enrolling in this study.”

3. A Supporting Information section is added at the end of our manuscript, line 333.

4. The original uncropped, unadjusted gel/blot images are compiled in a single PDF document named S1_raw_images.pdf. This file is uploaded as Supporting Information. The caption is now included in the Supporting Information section, line 334. This information is also provided in the Results section, lines 203-205.

5. Our figures were previously uploaded to PACE and those figures were already submitted. A revised Fig 4B is PACE cleared and uploaded.

6. Our datasets were uploaded to Dryad and documented in the manuscript in the Data Availability Statement beginning on line 335. Rovin, Richard (Forthcoming 2024). Soluble Axl Biomarker Datasets [Dataset]. Dryad. https://doi.org/10.5061/dryad.wpzgmsbwp. 

 Reviewer 1 specific responses

1. Gas6 measurments.

a. Response: We agree measuring Gas6 levels would be worthwhile. Unfortunately, the hospital closed our lab. We do not have access to specimens nor facilities to perform the analysis. We would also need to submit a protocol amendment to the IRBs. It is not clear if the IRBs would want another consent. If so, we would be unable to obtain consent form the GBM cohort as all have perished. 

2. sAxl levels in CSF. 

a. Response: This is interesting. However, our goal was to identify a circulating biomarker for GBM tumor burden. CSF obtained through lumbar puncture is not suitable for longitudinal monitoring of GBM patients.

3. Expand Table 3 to incorporate data from additional sources. 

a. Response: Thank you for this suggestion. Table 3 (now Table 4, see below) has been updated to include data from the hepatocellular carcinoma papers.[1–3] We were excited to review the Gustafsson paper regarding renal cell carcinoma.[4] Interestingly, renal cell carcinoma patients had lower levels of sAxl compared to controls with benign kidney disease. Though this was presented graphically, we could not find actual ng/ml values in the paper nor in the supplementary material. So, we were not able to include this in the table. 

4. Line 95. “1019”. 

a. Response: Thank you for catching this. The date is corrected to 2019.

5. Line 187, line 188, line 191, line 219. The correlation of between magnitudes should be quantified and included in the results.

a. Response: 

i. The Statistical analysis section is updated to included Spearman correlation coefficient determination beginning on line 167.

ii. The specific Spearman correlation coefficients, number of observations, and significance are summarized in a new Table 2. The existing Tables 2 and 3 are renumbered as Table 3 and Table 4. 

6. The values of the qRT-PCR analysis should be included in the study, not only the bar. The type of magnitude should be included in the figure legend (mean or median, SEM or SD) with the number of measurements performed.

a. Response: this oversight has been remedied. Table 3 is revised (line 213) and includes the actual Axl level as determined by qRT-PCR The legend for Fig 4B line 210) has been updated to include the magnitude and number of measurements.

Reviewer 2 specific responses

1. Results – It would be helpful to the reader if the Figures can be improved with high quality and pixels.

a. Response: the figures were uploaded to PACE to meet formatting requirements.

2. Tables-- The table looks a little bit complicated, just have a try the three-line table.

a. Response: We updated the tables to a cleaner, less cluttered look. Thank you for the suggestion.

3. Does this imply that the experimental results may be meaningless as there is no difference in preoperative AXL levels between the control group and the GBM group?

a. Response: Though our experimental results did not support our hypothesis, we would emphatically not characterize them as “meaningless”. We think such a suggestion is unusual. We, and others, believe that data contrary to expectations are often more interesting as they compel additional questions and experimentation.

4. By searching the keyword “glioblastoma” and “AXL”, it is not hard to find other articles on the connection between them. Please describe the difference between this manuscript and other articles and point out the innovations.

a. Response: We did perform a PubMed search using the terms “AXL” and “glioblastoma”. This returned 60 papers, 48 of which were relevant. Of these 48, 20 were related to the Axl signaling pathway, 19 looked at Axl as a treatment target, 6 were related to Zika virus, and 3 were biomarker related. One of these was the Hutterer et al., 2008 paper which is already discussed in our manuscript.[5] Another paper looked at phospho-Axl, the activated form of Axl, in glioblastoma tissue using immunohistochemistry.[6] The authors found that phosphor-Axl expression in both tumor vasculature and hypercellular areas was associated with a significant decrease in overall survival. Though important, this method is not suitable as a longitudinal measure of tumor burden. The third paper used a NanoString technique to determine differential mRNA expression of multiple genes in IDH wildtype and IDH mutant glioblastoma.[7] Axl expression was higher in the IDH mutant group. Again, an interesting technique and an important finding, but not suitable for longitudinal assessment of tumor burden. 

5. The language and syntax and format need more attention and the authors might benefit from the assistance of an English language editor.

a. Response: As a born English speaker, I am puzzled by this comment. Both Reviewers 1 and 2 affirmed that “the manuscript presented in an intelligible fashion and written in standard English”. If there are specific instances of poor word choice or convoluted sentence structure, please let us know so that we may address them. 

6. Suggestions

a. Use high quality pictures

i. Response: as above.

b. Identify innovative points of the article

i. Response: the innovative points of our study include the additional comparison of pre- and post- operative sAxl levels in patients with GBM. Moreover, we correlated volumetric MRI tumor metrics with sAxl levels. This can be found in our Conclusion section, lines 256-257. 

c. Expand the sample size and re-study to reduce false-negative results.

i. Response: this would be ideal. Unfortunately, we are unable to reopen this study and accrue additional patients.

d. Some of the references cited in the article are outdated.

i. Response: The oldest reference is from 1977. That paper is a classic in the Neurosurgical literature and it is still appropriate to cite. 

References

1. Song X, Wu A, Ding Z, Liang S, Zhang C. Soluble Axl Is a Novel Diagnostic Biomarker of Hepatocellular Carcinoma in Chinese Patients with Chronic Hepatitis B Virus Infection. Cancer Res Treat. 2020;52: 789–797. doi:10.4143/crt.2019.749

2. Dengler M, Staufer K, Huber H, Stauber R, Weiss KH, Starlinger P, et al. Soluble Axl is an accurate biomarker of cirrhosis and hepatocellular carcinoma development: results from a large scale multicenter analysis. 2017;8: 46234–46248. 

3. Reichl P, Fang M, Starlinger P, Staufer K, Nenutil R, Muller P, et al. Multicenter analysis of soluble Axl reveals diagnostic value for very early stage hepatocellular carcinoma. 2015. doi:10.1002/ijc.29394

4. Gustafsson A, Martuszewska D, Johansson M, Ekman C, Hafizi S, Ljungberg B, et al. Differential expression of Axl and Gas6 in renal cell carcinoma reflecting tumor advancement and survival. Clinical Cancer Research. 2009;15: 4742–4749. doi:10.1158/1078-0432.CCR-08-2514

5. Hutterer M, Knyazev P, Abate A, Reschke M, Maier H, Stefanova N, et al. Axl and growth arrest-specific gene 6 are frequently overexpressed in human gliomas and predict poor prognosis in patients with glioblastoma multiforme. Clinical Cancer Research. 2008;14: 130–138. doi:10.1158/1078-0432.CCR-07-0862

6. Onken J, Vajkoczy P, Torka R, Hempt C, Patsouris V, Heppner FL, et al. Phospho-AXL is widely expressed in glioblastoma and associated with significant shorter overall survival. Oncotarget. 2017. Available: www.impactjournals.com/oncotarget/

7. Zhang M, Pan Y, Qi X, Liu Y, Dong R, Zheng D, et al. Identification of New Biomarkers Associated With IDH Mutation and Prognosis in Astrocytic Tumors Using NanoString nCounter Analysis System. 2016. Available: www.appliedimmunohist.com

Very Sincerely Yours,

Richard A. Rovin, MD

Attending Neurosurgeon

---

## [Decision Letter · Decision Letter 1]

12 Jun 2024

Evaluating soluble Axl as a biomarker for glioblastoma: a pilot study

PONE-D-24-10611R1

Dear Dr. Rovin,

We’re pleased to inform you that your manuscript has been judged scientifically suitable for publication and will be formally accepted for publication once it meets all outstanding technical requirements.

Kind regards,

Salvatore V Pizzo

Academic Editor

PLOS ONE

Additional Editor Comments (optional):

Reviewers' comments:

Reviewer's Responses to Questions

**Comments to the Author**

1. If the authors have adequately addressed your comments raised in a previous round of review and you feel that this manuscript is now acceptable for publication, you may indicate that here to bypass the “Comments to the Author” section, enter your conflict of interest statement in the “Confidential to Editor” section, and submit your "Accept" recommendation.

Reviewer #1: All comments have been addressed

Reviewer #2: All comments have been addressed

2. Is the manuscript technically sound, and do the data support the conclusions?

Reviewer #1: Yes

Reviewer #2: Yes

3. Has the statistical analysis been performed appropriately and rigorously? 

Reviewer #1: Yes

Reviewer #2: Yes

4. Have the authors made all data underlying the findings in their manuscript fully available?

Reviewer #1: No

Reviewer #2: Yes

5. Is the manuscript presented in an intelligible fashion and written in standard English?

Reviewer #1: Yes

Reviewer #2: Yes

6. Review Comments to the Author

Reviewer #1: The authors have addressed my comments. I could not access the data in dryad, possibly the data are not yet available?

Reviewer #2: (No Response)

7. PLOS authors have the option to publish the peer review history of their article (what does this mean?). If published, this will include your full peer review and any attached files.

Reviewer #1: No

Reviewer #2: No

---

## [Editor Report · Acceptance letter]

25 Jun 2024

PONE-D-24-10611R1 

PLOS ONE

Dear Dr. Rovin, 

I'm pleased to inform you that your manuscript has been deemed suitable for publication in PLOS ONE. Congratulations! Your manuscript is now being handed over to our production team.

Kind regards, 

on behalf of

Dr. Salvatore V Pizzo 

Academic Editor

PLOS ONE